# Structural and Biophysical Characterization of the HCV E1E2 Heterodimer for Vaccine Development

**DOI:** 10.3390/v13061027

**Published:** 2021-05-29

**Authors:** Eric A. Toth, Andrezza Chagas, Brian G. Pierce, Thomas R. Fuerst

**Affiliations:** 1Institute for Bioscience and Biotechnology Research, University of Maryland, Rockville, MD 20850, USA; eatoth@umd.edu (E.A.T.); azzapium@gmail.com (A.C.); pierce@umd.edu (B.G.P.); 2Department of Cell Biology and Molecular Genetics, University of Maryland, College Park, MD 20742, USA

**Keywords:** hepatitis C virus (HCV), envelope glycoproteins, E1, E2, E1E2 glycoprotein complex, secreted E1E2, protein expression, protein purification, biophysical characterization, vaccine design

## Abstract

An effective vaccine for the hepatitis C virus (HCV) is a major unmet medical and public health need, and it requires an antigen that elicits immune responses to multiple key conserved epitopes. Decades of research have generated a number of vaccine candidates; based on these data and research through clinical development, a vaccine antigen based on the E1E2 glycoprotein complex appears to be the best choice. One bottleneck in the development of an E1E2-based vaccine is that the antigen is challenging to produce in large quantities and at high levels of purity and antigenic/functional integrity. This review describes the production and characterization of E1E2-based vaccine antigens, both membrane-associated and a novel secreted form of E1E2, with a particular emphasis on the major challenges facing the field and how those challenges can be addressed.

## 1. Introduction

The global burden of hepatitis C virus (HCV) infection is about 71 million or approximately 1% of the world’s population, with an annual rate of 1.75 million new infections each year [1]. Chronic HCV infection can lead to cirrhosis and hepatocellular carcinoma, which is the most common type of primary liver cancer [2]. The development of direct-acting antivirals has improved treatment options considerably, but their high cost restricts access, particularly in developing countries where the disease burden is greatest. Moreover, other factors such as viral resistance, the occurrence of reinfections after treatment cessation, and the fact that HCV is largely asymptomatic until the infected person sustains significant liver damage can diminish the impact of antiviral treatments. Therefore, the development of an effective preventative vaccine for HCV is imperative to lessen the burden of infection and transmission and to eliminate HCV globally [3].

Cumulative evidence has established that both B- and T-cell immunity contribute to the control of acute HCV infection [4,5,6]. In particular, broadly neutralizing antibodies (bnAbs) infused into humanized mice or chimpanzees protect against HCV infection [7,8,9]. Moreover, viral clearance is associated with a robust induction of bnAbs early in the infection [10,11,12,13]. Any protective HCV vaccine will likely require an immunogen that robustly elicits bnAbs. A major challenge is a high variability in the viral envelope E1E2 glycoprotein complex, the natural target of protective antibodies [14,15]. An ideal vaccine should elicit sufficient titers of bnAbs to multiple conserved E1E2 epitopes [8,16,17,18,19], in conjunction with cytotoxic and tissue-resident memory T-cells to achieve immunity, ideally sterilizing immunity, and protection against a high diversity of HCV isolates.

### 1.1. Importance of E1, E2, and E1E2 Epitopes In Virus Neutralization

In developing a prophylactic vaccine for HCV, the choice of antigen will be critical for its success. The HCV envelope glycoprotein complex, as the target of the protective antibody response, is currently the prime candidate. The envelope of HCV contains two glycoproteins, E1 and E2, that are encoded as part of the HCV polyprotein expressed in infected liver cells. This polyprotein is processed in the endoplasmic reticulum (ER) by signal peptidases and cellular glycosylation machinery to produce the mature E1E2 complex. These glycoproteins are membrane-anchored via their C-terminal transmembrane domains (TMDs), resulting in a membrane-bound E1E2 (mbE1E2) complex. A soluble version of the E2 protein (sE2) lacking the transmembrane domain [20] has been the focus of a number of vaccine studies [21,22,23,24,25,26,27]. However, immunological assessment in chimpanzees of an E1E2 vaccine produced superior immune responses compared to E2 administered alone and resulted in sterilizing immunity against a homologous virus challenge [28,29] but with some cross-neutralization capacity against heterologous isolates [30]. Moreover, E1E2 has been tested in humans and is well-tolerated [31]. However, due to the limited neutralization breadth observed in the human clinical trial [32,33], ongoing research efforts are now focused on developing an optimized, modified form of E1E2 (mE1E2) that can elicit more robust, cross-neutralizing responses against multiple heterologous isolates [25,34,35,36].

To develop a vaccine for HCV, the candidate antigen needs to be produced at a large scale, reproducibly, and at a cost that is not prohibitive. Most antibodies with broad virus-neutralizing (Vn) activities to diverse HCV isolates recognize conformational epitopes in E2 and some in E1E2 [8,37,38,39,40,41,42,43]. In addition, studies with a panel of E1-specific mAbs have shown that E1-specific responses can both protect against virus challenge and broadly neutralize HCV pseudoparticles (HCVpp) of various genotypes [44,45]. Previous work has identified highly conserved antigenic regions on E2 that elicit bnAbs [43,46,47]. These regions are called antigenic domains B, D, and E (used for this review). Other groups have designated such regions as epitopes I–III [48] or antigenic regions (ARs, which include epitopes that contain residues from both E1 and E2 [8,39]). Domains B and D are conformation-sensitive, whereas domain E contains a linear epitope [46]. However, while not conformation-sensitive in the same sense as domain B and D NAbs, domain E adopts multiple conformations, as observed when bound to different anti-domain E mAbs [49,50,51,52].

### 1.2. Structural Characterization of E2 Antigens and a Model of the E1E2 Heterodimer

Limited high-resolution structural data are available for the E1 glycoprotein. The E1 ectodomain contains 158 residues (residues 192–349 of the H77 sequence; NCBI Refseq ID NC_004102), of which 8 are cysteine residues that form 4 putative disulfide bonds and 5 are N-glycosylation sites. The reported X-ray structure of an N-terminal fragment of E1 (residues 192–270) exhibits “unexpected” domain-swapped and disulfide cross-linked oligomers [53], and it is a possibility that the observed structure and oligomerization were influenced by the C-terminal residue truncation, lack of E2 context, or both. As noted in a recent review [54], smaller, primarily helical fragments of E1 have also been structurally characterized (Figure 1).

The E2 glycoprotein has been structurally characterized more extensively than the E1 glycoprotein. The initially reported structures of the core E2 ectodomain structure [35,55] revealed a globular architecture stabilized by numerous disulfide bonds, which is expected due to the 18 cysteine residues present in the E2 ectodomain. Of note, different disulfide bonding patterns were observed in the two E2 core structures, possibly reflective of disulfide bonding heterogeneity on the native viral envelope or different truncations or source genotypes of the E2 sequences used in the two structures [56]. Subsequent studies have reported structures of E2 in complex with various monoclonal antibodies from humans [57,58,59,60] and, recently, macaques [61]. These include antibodies that engage the “neutralizing face” of E2 (Figure 1) that is used to bind the CD81 receptor, and while they exhibit different engagement strategies, many highlight the structural and mechanistic basis of shared features, such as biased VH1-69 germline gene usage and a CDRH3 loop disulfide motif [57,58]. Recent studies have provided insights into the flexibility and dynamics of the E2 neutralizing face through experimental characterization and computational modeling [36,62,63], suggesting possible “closed” and “open” E2 (or E1E2) conformational states associated with coreceptor binding and antibody neutralization. Analogous states and metastable behavior have been confirmed for other viral envelope glycoproteins, including HIV Env [64] and the SARS-CoV-2 spike [65].

The structure of the E1E2 glycoprotein, which would provide key insights into HCV and its immune recognition and enable structure-based vaccine design efforts, has not been experimentally determined to date. To address this need, independent studies have proposed multiple models of the E1E2 structure [66,67]; however, the lack of agreement between the E1E2 models [54], as well as inconsistencies between the initial computational models of E2 [68,69] and the E2 structure reported later, highlights the need for additional computational modeling of E1E2 in the absence of a high-resolution experimentally determined structure of this assembly. Advanced computational approaches such as the recently described artificial intelligence-based structure prediction algorithm, AlphaFold2, which was highly successful in a recent CASP13 structure prediction experiment [70], represent a promising avenue for accurate computational E1E2 modeling.

Structural data are essential for informing vaccine design, and the lack of E1E2 structural data represents a major deficit in the HCV vaccine development field. These data will aid in the optimization of neutralizing epitope presentation, engineering the antigen to make it more uniform, and enhancing stability. E1E2 is a complex antigen, so a rigorous biophysical analysis is an important part of the workflow for determining the optimal candidate. In this review, we describe how an E1E2-based vaccine can be produced and evaluated to ensure robust immunogenicity.

## 2. Expression Systems and Purification Considerations

The E1E2 complex is typically expressed as a polyprotein (Figure 2, top). This polyprotein is processed in the endoplasmic reticulum (ER) by signal peptidases and cellular glycosylation machinery to produce the mature complex (a trimer of E1E2 heterodimers) in a manner that mirrors HCV processing (Figure 2, bottom). The E1E2 complex has been difficult to prepare in the quantities needed for biochemical and immunological studies. Yields from the earliest large-scale preparations were quite poor (around 12 μg per liter of cell culture) [28]. Recently, significant progress has been made in the production and purification of the E1E2 complex via immunoaffinity purification [25,71] or the use of tags that allow protein A [72] or anti-Flag [73] chromatography. With purified and immunogenic antigen available, the next important milestone to facilitate vaccine development entails scaling up, selecting an expression host, and establishing a purification regimen that is suitable for biomanufacturing at a clinical scale while retaining the quality of the laboratory-scale product.

### 2.1. Expression Hosts

It is important to leverage robust laboratory-scale production of mbE1E2 to transition its production to systems that are suitable for the eventual clinical-scale manufacturing of vaccine antigens. Due to the extensive glycosylation of mbE1E2, bacterial expression hosts are not suitable for the production of this antigen. Based on its extensive track record with the Food and Drug administration (FDA), a preferred biomanufacturing system might seem to be Chinese hamster ovary (CHO) cells. However, there has not yet been a systematic examination of the relative quality, both as an antigen and an immunogen, of mbE1E2 produced using human embryonic kidney (HEK) 293 cells, CHO cells, and the baculovirus insect cell system (BICS). Each of these systems has been used in the past to produce mbE1E2, and they have both advantages and disadvantages, as described below.

#### 2.1.1. CHO Cell Expression Systems

CHO cells are the most widely used eukaryotic expression system for biomanufacturing, with an FDA approval history that spans over 30 years and over 50 approvals [75]. The glycan processing machinery, while not human [76], is similar to human machinery, so the glycosylation pattern of a CHO-derived antigen is similar to that of an antigen produced during a natural HCV infection. CHO cells grow to high cell density in serum-free media and are amenable to continuous production processes. Moreover, cGMP cell banks of CHO cells are readily available, and commercial feeds and additives have been developed for cGMP processes with CHO cells. The CHO system has been used for nearly 30 years to successfully produce mbE1E2 [72,77]. The CHO expression system used to produce the E1E2 antigen that entered clinical studies [32] employed a stably transfected cell line (for native mbE1E2 [77]); recently, a lentiviral transduction system for Fc-tagged mbE1E2 has been used. Current yields for both systems are approximately 1 mg of purified mbE1E2 per 100 g of CHO cells [72].

#### 2.1.2. HEK 293 Cell Expression Systems

HEK 293 cells are a common host for producing recombinant proteins for research. Their FDA portfolio is somewhat limited, with five therapeutic agents approved by the FDA since 2011 [75]. This includes a treatment for septic shock, three coagulation factors used in the treatment of hemophilia, and a glucagon-1-liker peptide for the treatment of diabetes [75]. One key advantage of the HEK-based system is that, since the cells originate from a human embryonic kidney [78], the post-translational modification (PTM) machinery is human. Each of the approved therapeutic agents produced in HEK 293 cells requires a PTM that is not efficiently produced in other systems [79,80]. This PTM requirement is particularly relevant for HCV vaccine development as HEK-produced E1E2 will have a glycan content containing mostly complex- and high-mannose-type glycans, which likely resembles the E1E2 glycan content observed in a natural HCV infection. Our original transient transfection used the freestyle HEK 293 system and a pcDNA3.1 (+) vector containing a CMV promoter to produce highly purified immunogenic mbE1E2 antigens on a laboratory scale. While this system worked well for small-scale biochemical and immunological assessments, it was insufficient for larger studies due to low yields (100 to 300 μg per liter of cell culture, unpublished data). We, therefore, sought to increase yield by focusing on the influence of the signal peptide used and maximizing host cell density during production runs. Our original system employed the natural HCV signal peptide, which contains the C-terminal 22 residues of the HCV core protein. However, we and others [81] have shown that alternate signal peptides can increase yields of secreted forms of E2 containing just the ectodomain (sE2). We tested the effect of different signal peptides, such as the signal peptide from tissue plasminogen activator (tPA), on the amount of mbE1E2 produced. We also examined the use of Expi293F cells, which are suspension HEK 293 cells conditioned to grow at high density and with increased protein production, as an additional means to increase yields. Our data (Figure 3) does, in fact, show a marked increase in the E1E2 produced. The samples in Figure 3 are normalized for cell weight per mL of culture, and Expi293F cells yielded 3-fold more E1E2. Accounting for the 3-fold increase in cell weight per mL of culture results in a total increase of 9-fold more E1E2 with eExpi293F cells. Current yields from this system are 2 to 5 mg per liter of Expi293F cells.

#### 2.1.3. BICS Expression Systems

The BICS system has a good track record in viral vaccine development, with two commercial viral vaccines approved for use in humans (Cervarix™ and FluBlok™) [82] and five viral vaccines approved for use in veterinary medicine. The strengths of BICS stem from 35 years of research and development since its original description in 1983 [83], which has contributed to its optimization. One clear strength of BICS is the ease of scale-up, which has now reached 21,000 L. Another strength is the availability of sophisticated tools and approaches for manipulating baculoviruses. Among these tools are “bacmids”, which are modified baculovirus genomes cloned into a bacterial plasmid that can be used to quickly and easily create recombinant baculoviruses encoding multiple heterologous gene products. Work on modifying protein processing pathways in BICS has been ongoing for over 25 years, making BICS a candidate system for cell line engineering. Finally, previous studies have shown the two insect cell lines most commonly used as hosts in BICS, which are derived from *Spodoptera frugiperda* (Sf; e.g., Sf21 [84], Sf9 [85]) or *Trichoplusia ni* (Tn; Tn368 [86], BTI-Tn-5B1-4, commercialized as High Five™ [87]) are contaminated with adventitious viral agents [88,89]. Thus, new Sf- and Tn-derived cell lines that lack any adventitious viral agents were recently isolated. These new cell lines are designated Sf-RVN (for rhabdovirus-negative [90]) and Tn-NVN (for nodavirus-negative [91]), respectively. Extensive transcriptomic and genomic analyses have shown that both cell lines lack any known viral contaminants [92], making these systems ready for production up to the clinical scale. Despite the above advantages, BICS is rarely used for the expression of mbE1E2, although one report [93] has described the expression and purification of mbE1E2 from Sf9 cells. The purified mbE1E2 complex was competent to bind the receptor CD81, indicating structural and functional integrity.

### 2.2. Purification of mbE1E2

In order to advance mbE1E2 purification for vaccine development, any purification regimen must produce highly purified, antigenically intact glycoprotein using methods that can be scaled up for eventual clinical use. This has been a significant challenge in the field of HCV vaccine development. For initial vaccine trials using mbE1E2, purification entailed using agarose-coupled lectin from *Galanthus nivalis* (GNA) as the capture step [94]. While this method significantly enriches mbE1E2 relative to the solubilized membrane fraction, any host cell protein containing glycans with mannose will bind to the resin. This lack of specificity, as well as potential influences of expression hosts on the glycan content of both mbE1E2 and host cell proteins, makes GNA unsuitable as an initial capture step at large scales [95,96]. Therefore, successful utilization of purification methods with a higher degree of specificity and scale-up capability would represent a significant advance in HCV vaccine development. We will discuss some recent advances in producing purified mbE1E2 using various affinity purification strategies.

One recent successfully employed purification strategy entails inserting the Fc domain from IgG1 and a protease cleavage site between the E1 and E2 open reading frames in the E1E2 polyprotein (i.e., such that it is fused to the N-terminus of E2) [72]. The authors reported that more standard affinity tags, such as hexahistidine and twin-Strep, were unsuccessful in enriching mbE1E2, presumably due to a lack of tag accessibility to the resin. The Fc-tag strategy allows protein A or protein G purification, both of which are suitable for scale-up. In this strategy, membrane extracts from CHO cells expressing mbE1E2 are chromatographed using a commercially available protein G sepharose column. The reported method uses the column cleavage with commercially available rhinovirus 3C protease, followed by removal of the protease via glutathione-sepharose chromatography and passive purification with hydroxyapatite chromatography [72]. The resulting product is highly purified, formed E1E2 heterodimers that are both antigenically and immunogenically intact [72]. However, there are risks that must be considered when using such a method. In particular, the tag is large, and it is a potentially immunogenic tag. Therefore, it must be efficiently cleaved, and all traces of the tag must be removed following purification, either through additional chromatography steps or some other means. Moreover, if on-the-column cleavage is inefficient, harsh elution protocols (low pH or high concentrations of chaotropes) are required following capture.

An additional affinity chromatography strategy involves placing a flag-tag immediately before domain E of the E2 ectodomain, taking advantage of a flexible, highly exposed region of the glycoprotein to facilitate affinity capture. The published method uses anti-flag affinity chromatography as a single chromatographic step, and elution of the bound flag-mbE1E2 is achieved by incubating the resin for a limited time in glycine pH 3.5 [73]. As with the Fc-tag strategy, the flag-mbE1E2 produced is antigenically and immunogenically intact. While the anti-flag resin can be regenerated a number of times, it is not clear whether this method can be efficiently scaled up for vaccine production.

Immunoaffinity chromatography (IAC) is also a method that can be used to isolate highly purified mbE1E2. Similar to the anti-flag strategy, IAC is highly specific, with the added feature of selecting for desirable properties in the purified antigen, such as conformationally intact epitopes known to be important for eliciting bnAbs [97]. This technique was first employed using the antibody H50 coupled to activated CNBr sepharose [71]. In this initial report, IAC was the initial capture step and was followed by GNA affinity chromatography. The mbE1E2 purified using this regimen is able to bind its coreceptor CD81 and can be reconstituted into liposomes. However, the elution conditions are harsh (pH 12), and, in our hands, direct application of solubilized membrane fractions onto the antibody column significantly limited its useful lifetime due to clogging (unpublished results). Our group has expanded on this IAC strategy to include initial cleanup steps using ion-exchange chromatography. First, anion exchange is used to remove lipids, nucleic acids, and aggregated mbE1E2 present in the membrane extract. Next, a cation exchange step is used to remove additional aggregated material. In both of these steps, the impurities bind to the column matrix, and mbE1E2 is present in the flow-through. The flow-through from cation exchange is then applied to the IAC column. We chose the conformation-sensitive mAb HC84.26.WH.5DL [7,43] for this step owing to the importance of its binding epitope domain D in eliciting bnAbs. Elution is accomplished with a chaotrope, sodium thiocyanate, which is immediately dialyzed away from the purified sample. Should impurities exist after this step, further purification is accomplished with GNA affinity chromatography. The mbE1E2 produced in this manner is, as in the other methods, highly purified (Figure 3) and both antigenically and immunogenically intact [22,98]. The main drawback is the need to make antibody-coupled resin by hand as this material is not commercially available. Whether such a strategy can be applied at scale is unclear.

While these three methods all produce high-quality samples, they all involve harsh elution conditions. How such conditions might influence sample quality at large scales is unclear. Further, all three methods involve intracellular expression and membrane extraction. This limits the ability to produce large quantities of sufficient homogeneity required for both basic research and vaccine production. Development of the soluble native envelope glycoproteins SOSIP (for HIV) [99], DS-Cav1 (for RSV) [100], and the “2P” variant of the MERS spike ectodomain [101] greatly accelerated vaccine research efforts for these diseases. Unfortunately, similar advances in the HCV field have been slower to develop. Liberating the complex from the membrane in its native form would allow rigorous determination of the biochemical properties and intermolecular interfaces that define a native E1E2 complex. We discuss recent advances in this area at the end of the article.

## 3. Biophysical Characterization of E1E2

Developing a strategy to elicit broadly protective neutralizing antibodies is essentially an immunological “signal-to-noise” problem. The “signal” is the effective presentation of neutralizing epitopes to the immune system, and the “noise” is all of the deleterious properties of a complex immunogen like E1E2, i.e., immunodominant decoy epitopes, sample heterogeneity, low yields, poor solution behavior, and instability. As part of the validation process, it is important to define the desirable properties of an E1E2-based vaccine antigen and develop a rigorous set of biophysical assays to assess those properties. A necessary precursor to such an endeavor is basic research on the assembly and properties of E1E2 complexes, both in solution and in the virion. In this section, we will review what has been done to experimentally describe a native-like E1E2 and how to assess native-like properties when producing candidates for vaccine studies.

### 3.1. Size, Homogeneity, and Oligomeric State

While the basic functional unit of the HCV glycoprotein complex, a heterodimer of E1 and E2, has been known for some time [94,102,103,104], the organization of the E1E2 complex in the context of viral particles, i.e., a trimer of heterodimers, has been poorly understood until recently, although this arrangement had been previously proposed [94]. Moreover, biochemical characterization of mbE1E2 proved difficult, in part, due to the fact that that coexpressed E1 and E2 ectodomains alone do not form a stable complex [105]. One study did identify a series of mutations in E1 that impair heterodimerization [106]. Investigation of the determinants for mbE1E2 complex formation, however, has focused more heavily on the non-ectodomain regions, namely, the stem and transmembrane domains. A portion of the E2 stem (675–699) contains some residues that impact E1E2 heterodimerization [107], whereas the remainder of the E2 stem [108] and the entire E1 stem (330–347) do not contain residues important for heterodimerization [109]. The structural elements that appear to provide the main driving force for heterodimerization are the E1 and E2 TMDs [110]. A series of studies employing alanine insertion mutagenesis, replacement of charged residues with hydrophobic residues, and tryptophan mutagenesis [98,111,112] identified important residues within the E1 and E2 TMDs that are important for heterodimerization, including a GXXXG motif in E1 [112]. As this motif is known to promote oligomerization [113,114], the above series of studies led to additional attempts to define the oligomeric state of E1E2 in viral particles. This analysis was complicated in part by the fact that E1E2 forms large complexes on the surface of the virion that are stabilized by intermolecular disulfide bonds [115]. In a series of biochemical experiments [116] using DTT to disrupt intermolecular disulfides, the authors showed that E1E2 exists as a trimer on the surface of the virion. In particular, by fractionating particles in a sucrose gradient and using Western blotting with gentle heating (i.e., 37 °C) to analyze the fractions containing viral particles, they observed that the trimeric E1 protein is present in preparations of both cell-cultured HCV (HCVcc) and HCVpp. Moreover, the authors showed that the E1 TMD drives trimer formation as trimeric species of E1, but not E2, could readily be detected in HCVcc and HCVpp. Trimerization appears to be an inherent property of the E1 TMD, as this domain, when fused to the normally monomeric bacterial thioredoxin, induces trimer formation. Based on their mutagenesis experiments, this trimerization propensity requires an intact GXXXG motif. Interestingly, despite this property of the E1 TMD, no trimer was observed in HCVpp containing E1 alone. This suggests that E2 plays a critical role in assisting E1 to fold into its native state, which is presumably required to facilitate the transition from monomer to trimer. Purified mbE1E2 also exhibits E1 trimer formation (Figure 4). In light of the above observations, any E1E2 vaccine candidate would, by necessity, need to exist in one of the observed quaternary states, i.e., heterodimer or trimer of heterodimers, to be optimally effective. Ideally, the vaccine candidate would be uniformly either the heterodimer or trimer of heterodimers, as such a preparation would be more reproducible from lot to lot, whereas mixed oligomeric populations are more likely to have unacceptable variability. Below, we describe the associated results for three methods that can be used to describe the oligomeric state and heterogeneity present in mbE1E2 preparations: size exclusion chromatography (SEC), analytical ultracentrifugation (AUC), and SEC combined with multiangle light scattering (SEC-MALS).

#### 3.1.1. SEC

SEC, as the name suggests, fractionates protein samples by size based on the exclusion of proteins from the pores of beads that comprise the column matrix. This technique can be used for purification on the preparative scale or for analytical purposes, e.g., estimating the molecular weight of a given sample based on its retention time relative to standards of known molecular weight. Shown in Figure 5 is an SEC profile and the associated Western blots using purified mbE1E2 that was expressed in Expi293 cells. In the chromatogram, purified mbE1E2 elutes at a molecular weight consistent with a very large protein species (>300 kDa). The predicted molecular weight of the E1E2 trimer of heterodimers is 270 kDa, based on the protein molecular weight (180 kDa) plus 45 glycans each, with an average molecular weight of 2 kDa [117] per glycan. Two factors that are relevant to mbE1E2 influence its analysis by SEC. First, SEC is highly dependent on the shape of a given protein. Asymmetric proteins have a larger effective hydrodynamic radius than spherical proteins of the same molecular weight [118], and, thus, they elute from SEC earlier than expected based on molecular weight. Current models of mbE1E2 predict an asymmetric shape [66,67] and, thus, an overestimate of molecular weight by SEC as the molecular weight standards are chosen to be approximately spherical, well-behaved proteins. Second, glycosylation increases the effective radius of a protein (even when accounting for the extra mass) [118], resulting in an additional overestimation of the apparent molecular weight. These two complicating factors make SEC inappropriate for determining the size of mbE1E2, i.e., beyond a general diagnostic assessment of lot-to-lot consistency. Analytical-scale SEC to assess consistency between mbE1E2 preparations can be a useful diagnostic step. Moreover, SEC does not allow fine enough separation in this molecular weight range to provide detailed information about heterogeneity. The peak in Figure 5 is broad, indicating heterogeneity, but additional experiments are required to provide granular detail on heterogeneity.

#### 3.1.2. AUC

In AUC, protein solutions are subjected to a strong centrifugal field, causing the species within those solutions to sediment in bands according to their molecular mass and shape [119]. Fitting the resulting data to the Lamm equation allows estimation of the sedimentation coefficients of the individual species in solution. This allows sensitive detection of different molecular mass species in solution, from aggregates to protein complexes to monomers of a given protein of interest. This makes AUC very useful for deconvoluting heterogeneous protein mixtures, thereby providing essential information about the solution behavior of a given vaccine candidate. When applied to mbE1E2 (Figure 6), this analysis reveals a highly heterogeneous mixture of protein species, with major peaks at 4.0 S, 6.6 S, and 9.2 S, plus a shoulder and minor species (presumably aggregates) in excess of 10 S [22]. This analysis is consistent with the breadth of the SEC profiles in Figure 5 and previous studies using glycerol gradients and Western blotting [94]. The scope of the issue of heterogeneity, as it pertains to vaccine development, is aptly illustrated by these data. Ideally, there should be a single species with a sedimentation coefficient consistent with a trimer of heterodimers. One of the major challenges, moving forward in developing a vaccine against HCV, will be producing a candidate that is reproducibly uniform in solution, and AUC appears to be a powerful diagnostic tool for evaluating such candidates.

#### 3.1.3. SEC-MALS

Light scattering depends strongly on the size of the particle involved, making it an excellent technique for determining molecular size. MALS is the preferred technique because multiangle scattering accounts for a nonspherical macromolecular shape [120]. This technique is coupled to SEC to allow the analysis of polydisperse or heterogeneous systems. In SEC-MALS, UV detection of the protein eluting from the SEC column is used to estimate the protein concentration for a given species, and light scatter is then used to determine the molecular weight, thereby creating a molecular mass profile for all resolved peaks in the chromatogram. As applied to mbE1E2 (Figure 7), we observed a broad peak centered about 1.1. MDa, containing species in size ranging from 500 to 2.5 MDa. This is consistent with both the SEC alone and AUC data, indicating that significant heterogeneity exists in these mbE1E2 preparations. The precision and ability to account for shape, afforded by SEC-MALS, makes this another powerful diagnostic tool for assessing size and heterogeneity.

The above techniques together comprise a thorough biophysical assessment of size and heterogeneity of E1E2 vaccine candidate preparations. However, the analysis is posthoc, and, thus, a major challenge in the field of E1E2-based vaccine development will be the production of homogenous E1E2 preparations. This will likely require both antigen and expression system engineering, so it will be instructive to consider some factors that contribute to solution heterogeneity in mbE1E2 preparations, as these will necessarily need to be controlled going forward. Below, we focus on disulfide bonds and glycosylation in E1E2.

#### 3.1.4. Factors Affecting Size and Homogeneity: Disulfides

The E1E2 complex contains a number of disulfide bonds, with four likely intramolecular disulfides in E1 and nine intramolecular disulfides in E2 [121,122]. Whether an intermolecular disulfide exists between E1 and E2 molecules in the heterodimer is unclear, based on putative structural models [66,67], but early biochemical evidence [94,102] suggests that if such a disulfide-bonded species exists, that species is a minor one. Appropriate disulfide bond formation is an integral part of E1E2 complex folding. Based on data from heterologous expression systems, the process is slow and requires the glycoprotein-specific chaperones calnexin and calrectulin [123,124] and likely involves protein disulfide isomerases such as ERp57 [125]. In aggregate, the data pertaining to E1E2 polyprotein processing point towards a system in which E1 and E2 mutually assist folding and assembly into a native trimer of heterodimers [116,123,124]. Conservation of these cysteines across E1E2 sequences from many genotypes indicates that the disulfides are essential [121,122]. However, the disulfide bonds in E1 do not appear to be essential for E1E2 complex formation as systematic mutation of the conserved cysteines in E1 does not impair heterodimer formation [122]. Rather, these mutations impaired viral particle assembly and rendered viral particles less stable than their wild-type counterparts [122]. By contrast, E2 disulfides appear to be functionally essential, as mutation of any one cysteine to alanine abolishes the infectivity of HCVpp. Moreover, all of the individual cysteine mutants exhibited some impairment of heterodimer formation. Only C503A and C620A mutants retained some ability to form E1E2 heterodimers. This phenomenon is not likely a simple consequence of the existence of free thiols in the final complex, as mutation of both C581 and C585, which form a disulfide bond, does not restore the E1E2 complex formation. [121]. However, not all these cysteines are essential for maintaining a conformation that is competent to bind antibodies as the E2 ectodomain can tolerate eight cysteine substitutions and still bind the monoclonal antibody H53 [121].

Early reports from heterologous expression systems pointed towards disulfides as a source of potential heterogeneity. A fraction of the protein expressed in mammalian cells using vaccinia and sindbis viruses formed non-native disulfide-crosslinked complexes [102]. A later report showed that E1E2 complexes in HCVcc form large disulfide-crosslinked species [115] that, despite their covalent tethering, are competent to bind both CD81 and conformation-sensitive mAbs. The authors speculated that these complexes could arise from disulfide-shuffling, suggesting that the malleability of the disulfide configuration in E1E2 is an inherent and perhaps functionally relevant property of the complex. Further data pointing towards disulfides as a source of heterogeneity have come from biochemical and structural work. In particular, a biochemical investigation using N-terminal sequencing and mass spectrometry mapped out the disulfide bond connectivity of the E2 ectodomain, thereby arriving at a configuration for all nine disulfides [69]. Subsequent crystal structures of truncated forms of the E2 ectodomain also provided a glimpse at disulfide connectivity [35,55]. Surprisingly, there was notable disagreement between these three studies. The two crystal structures shared only three disulfides in common (of a potential five, as the smaller construct only contained 11 cysteines). These data lend credence to the idea that disulfide bond rearrangement is an inherent propensity of E1E2. Later structural work on the complete E2 ectodomain [58] identified one arrangement comprising eight of the nine disulfide bonds (Figure 8). Of these, the biochemical studies identified two, and the third (C652-C677) is likely correct by process of elimination. Of the other six, at least three pairs (C452 and C459, C486 and C494, and C503 and C508) are close enough in the structures that they might be part of a bona fide alternate disulfide configuration. For the structural studies, a plausible explanation is that the truncations required for crystallization induced the formation of non-native or alternate disulfide bonds [56]. The rationale behind this effect could be that the disulfide bonds are required to stabilize the protein, and, thus, the structure adapted to satisfy that energetic requirement in whatever manner was possible. The E2 ectodomain exhibits a surprisingly high melting temperature [25,36] for a protein that contains greater than 50% flexible loops and unstructured regions. Stabilization via disulfide bonding is an adaptive mechanism with precedent in thermophilic proteins, particularly in extreme thermophiles (organisms that exist in environments at or above 100 °C) [126,127]. Perhaps the envelope glycoproteins evolved this unusual structure, deficient in secondary structural elements, and knitted together with disulfide bonds to maximize both stability and flexibility as a means to evade the immune system without compromising infectivity.

Our own observations indicate that inter-heterodimer crosslinking occurs in purified mbE1E2 preparations, analogous to what has been observed in HCVcc. In Figure 9, when comparing reducing versus nonreducing Western blots using anti-E2 mAb HCV-1, the reducing blot shows a single band, whereas the nonreducing sample shows a clear laddering of HCV-1 reactive bands extending nearly to the top of the gel. This is in stark contrast to the E2 ectodomain alone, which exhibits primarily monomeric E2 with a small fraction of dimer under the same conditions (not shown). Given this propensity of E1E2, a pressing question in the production of E1E2-based vaccine candidates is how to control or eliminate disulfide bond exchange and oligomerization/aggregation. From an antigen production perspective, the ideal vaccine candidate will be a reproducibly homogenous preparation, so eliminating disulfide-bond-exchange-induced heterogeneity would represent a major advance. There are three likely means to isolate E1E2 preparations that possess no inter-heterodimer crosslinks. The first is purification. Perhaps judicious chromatographic separations could isolate enough of such populations in E1E2 preparations to make a viable vaccine candidate. The second potential means of eliminating disulfide crosslinking is antigen engineering. As the E2 cysteines appear to be both required for heterodimer formation and the major drivers of crosslinking, such an effort would likely entail suppressing protein flexibility via mutagenesis to discourage the rearrangement of disulfides that allows inter-heterodimer crosslinking. The third potential way to eliminate disulfide bind crosslinking between E1E2 heterodimers is cell line engineering. It is possible that crosslinking occurs due to the presence of multiple different disulfide arrangements while the recombinant protein is being produced in eukaryotic expression systems. If this is the case, then altering the protein disulfide isomerase machinery in the cell lines used for expression might produce a single disulfide configuration with a low propensity for rearrangement and crosslinking.

#### 3.1.5. Factors Affecting Size and Homogeneity: Glycans

Both E1 and E2 are heavily glycosylated, with four conserved glycosylation sites in E1 and nine conserved sites in E2 (referred to as E1NX and E2NX, respectively). An additional site (E1N5) exists in E1 in some strains, and the sites E2N5 and E2N7 are lacking in some genotypes. Strain H77, which is frequently used for biochemical characterization and as an immunogen, contains 11 glycosylation sites in the E2 glycoprotein. Functionally, these glycans act as a shield, reducing access to neutralizing antibodies. In particular, one study found that glycans E2N1, E2N2, E2N4, E2N6, and E2N11 markedly reduce HCVcc sensitivity to neutralization by antibodies [128]. In addition, some glycans (E2N3, E2N7, and possibly others) appear to be important for viral entry, at least for the H77 strain [128,129].

From a vaccine production standpoint, minimizing any potential heterogeneity arising due to glycosylation will aid in the development of a reproducibly homogenous immunogen. Heterogeneity of this type can arise via two effects. The first is the natural variability in glycan species conferred by heterologous expression systems. In general, CHO and HEK systems append complex and high-mannose-type N-linked glycans [26,130], whereas BICS such as Sf9 appends less complex glycans [26]. These glycans tend to be quite variable both at a given site and across different sites in a given antigen [26]. The variation of glycan distributions from species to species does not appear to appreciably affect either antigenicity or immunogenicity of a given antigen, potentially because the variability itself essentially blurs out any differences that might be observed [26]. The question remains regarding how much these variable glycans contribute to heterogeneity in the solution. From a broad perspective, bands on SDS-PAGE and Western blots broaden. Curiously, this effect is, in some cases, more pronounced in secreted forms of E1E2 (or E2) than those that remain in cells [22]. However, from a biochemical perspective, it does not appear to be a major contributor, at least relative to disulfide bond crosslinking. For example, deglycosylated E1E2 complexes exhibit as much, if not more, heterogeneous banding patterns in nonreducing Western blots [22].

A second potential contributor to heterogeneity is variable glycan site occupancy in the antigens produced in heterologous expression systems. This issue has received little attention, but recent results from our group highlight why glycosylation site occupancy should be well-controlled, if possible. It has been established that deletion of some glycan sites such as E2N8 and E2N10 results in defective E1E2 complex assembly and lack of infectivity in either HCVcc or HCVpp [131] for the H77 strain. Other sites show infectivity defects, but their potential effects on E1E2 assembly are less clear. While this level of defect might not hold for all genotypes [132], the occupancy of these sites should be monitored, especially when H77 is the strain of choice for an E1E2 immunogen. A recent analysis of the E2 ectodomain produced in HEK 293 cells and Sf9 cells [26] demonstrated that the percentage of aglycosylated sites varied markedly depending on the glycosylation site within the E2 ectodomain and the expression host. In HEK 293 cells, sites E2N2 and E2N9 were greater than 99% glycosylated, while site E2N3 was only 75% glycosylated, and sites E2N10 and E2N11 were 48% and 28% glycosylated, respectively. Similar trends were observed for Sf9 cells, but the levels of glycosylation for sites E2N10 and E2N11 were lower (19% and 3%, respectively). A similar study of E1E2 constructs would be highly informative. Given the role of site E2N10 in E1E2 heterodimer formation and the fact that these glycans are in a position to shield domain A, which gives rise to non-neutralizing antibodies, these data show that glycan site occupancy should not be ignored. Moreover, unoccupied glycan sites could give rise to neoepitopes, as was the case for a glycan deletion construct of HIV Env [133]. In this case, strong neutralizing titers were obtained for deglycosylated viruses but not wild-type viruses. Finally, this type of heterogeneity is pernicious, i.e., if the antigens with aglycosylated sites are otherwise functional, removal from the population of E1E2 complexes would prove extremely difficult.

### 3.2. Antigenicity and Receptor Binding

#### 3.2.1. Antibody Binding

Given the importance of the antibody response in clearing HCV infections, optimal immunogens should exhibit robust binding to bnAbs. Antibodies that bind to E1, E2, and E1E2 have been extensively characterized, allowing a thorough examination of antigenicity. Many of these mAbs are neutralizing, and a subset of those bind to conformation-sensitive epitopes. Thus, antibody binding is an excellent tool to ensure the structural integrity of potential vaccine candidates, particularly when using methods that can establish a quantitative baseline for comparison. In particular, these antibodies can be used to probe the integrity of epitopes known to elicit bnAbs, as these epitopes will be a major determinant of the efficacy of an eventual vaccine. Moreover, efforts to stabilize important epitopes via the introduction of stabilizing mutants or shield epitopes known to elicit non-neutralizing antibodies can be first evaluated in vitro based on the affinity of an antigen for the appropriate mAb. In a recent study by our group [25], we identified a stabilizing mutation that enhances affinity for the domain D mAb HC84.26.WH.5DL in the context of the soluble E2 ectodomain. The same study confirmed that glycan insertion can inhibit binding to the non-neutralizing domain A mAb CBH-4G [25]. In addition to productive changes resulting from rational design efforts, antibody binding is a useful analytical tool for ensuring consistency from the lot-to-lot production of antigen. A marked decrease in affinity to one or a number of conformation-sensitive mAbs would be indicative of a poor-quality antigen caused by any number of factors (e.g., failed design, bad preparation, improper storage, shipping snafu) and could thereby prevent resources from being wasted on downstream experiments.

The major quantitative techniques used to assess antibody binding are surface plasmon resonance (SPR), biolayer interferometry (BLI), isothermal titration calorimetry (ITC), and enzyme-linked immunoassay (ELISA). ITC is label-free and requires no immobilization. Moreover, ITC can give thermodynamic and stoichiometric information [134]. Disadvantages of the technique are that several hundred micrograms of protein are typically required, the samples need to be highly purified, and high-affinity interactions (low nM and tighter) require displacement titration methods to accurately determine a dissociation constant [134]. SPR, BLI, and ELISA all entail immobilizing the antigen or antibody to a surface and incubating it with varying concentrations of the binding partner to derive estimates of the affinity of the binding reaction. In each technique, the ease and reproducibility of immobilization is a key consideration. Covalent immobilization is highly efficient but can result in a small fraction of the immobilized material retaining its functional and structural integrity. Immobilization techniques that are less prone to such effects, such as Ni^2+^-NTA (tris-nitrilotriacetic), anti-His, anti-Fc, and GNA, are also commonly used for proteins that contain hexahistidine tags, Fc-fusions, or glycans with mannose. SPR and BLI are kinetic techniques that can give information about the association and dissociation rate constants of the binding reaction [135,136]. These rate constants are then used to determine affinity. Kinetic methods are particularly useful in discriminating subtle differences in binding between mAbs, e.g., when the measured affinity is similar but there are marked differences in the kinetic rate constants. ELISA is a direct measure of binding affinity at equilibrium that requires several immobilization steps (e.g., antigen, primary mAb, secondary mAb) and an indirect readout, typically using horseradish peroxidase and a chromophore [137]. Of the above methods, ELISA is less commonly used for the determination of dissociation constants.

Regardless of which method is used to analyze antigenicity, the choice of antibodies used for evaluation will have a significant impact on the decision-making process for immunogen evaluation. Fortunately, the HCV field has collectively assembled a large compendium of human mAbs that recognize epitopes in E1, E2, and the E1E2 complex. Many of these mAbs have been characterized structurally, either in complex with peptides defining their epitopes [7,49,50,51,52,138,139,140,141,142,143,144,145,146,147,148,149,150,151] or the antigen itself, in the case of the E2 ectodomain [35,55,57,58,59,60]. In addition, there have been comprehensive epitope-mapping studies [152,153] that have further defined key residues that mediate antibody–antigen interactions. All of this information can be leveraged to assemble a set of antibodies to be used in evaluating candidate E1E2-based immunogens. To avoid missing differences in binding that would otherwise be obscured due to the precision of the antibody–antigen interaction, it is useful to measure binding to at least two different mAbs per antigenic site of interest. An example set of mAbs for this manner of evaluation is shown in Table 1. In two separate studies by our group, we identified differences in the binding affinity of two antibodies to the same antigenic domain in the same antigen. In the first study mentioned above, the stabilizing mutation that affects binding to the domain D mAb HC84.26.WH.5DL did not affect binding to the antibody HC84.1, which binds to the same domain [25]. In addition, the domain A shielding glycan affected binding to CBH-4G but not CBH-4D [25]. In the second study in the context of E1E2, binding of the antibodies AR4A and AR5A to a secreted scaffolded construct called sE1E2.LZ (described in detail below) differs by approximately nine-fold despite the similarity of the epitopes contacted by the two antibodies. It should be noted that AR4A and AR5A do not compete for binding to E1E2 [39], perhaps explaining the origin of this difference. Other groups have observed similar instances of differences in the binding of some pairs of antibodies to the same antigenic domain. In one instance, the binding of AR3C and AR3D [57] to a multimeric form of the E2 ectodomain exhibited an approximate eight-fold difference [21]. In addition, epitope mapping studies showed that for the W420A mutant, the binding of all HC84-family mAbs was robust except for HC84.22 and HC84.23, which exhibited a five- to eight-fold decrease in binding to mbE1E2, relative to the other HC84 mAbs [43]. These differences, while moderate, could impact immunogenicity and, thereby, influence design strategies. Inadequate antigenicity testing risks missing productive modifications on the path towards the optimal immunogen for vaccine studies.

#### 3.2.2. Receptor Binding

To facilitate viral entry, E1E2 interacts with host cell receptors. The minimal set of HCV coreceptors sufficient to allow viral entry [157] are the tetraspanin CD81 [158], scavenger receptor class B type I (SR-BI) [159], claudin-1 (CLDN1) [160], and occludin (OCLN) [161]. E1E2 coreceptor binding is a true functional test that examines the overall integrity of the antigen and is, therefore, a key diagnostic test that should precede downstream studies. In general, the large extracellular loop of CD81 (CD81-LEL) is used for analysis because this protein fragment is easy to produce and well-behaved. By contrast, cell-based assays are required to assess SR-BI interactions (e.g., [162]). A number of techniques have been used to assess binding to CD81, including cell-based assays [163], pulldowns [164], ELISA [93,165], and more quantitative methods such as ITC [36], SPR [22], and BLI [25,26,60]. As is the case for mAb binding, quantitative methods are preferred to give a more robust comparison. CD81 binds the neutralizing face of E2 [166], including residues from antigenic domains B, D, and E, domains which elicit bnAbs. Thus, wild-type levels of binding to CD81 indicate that these antigenic domains are properly configured and that the immunogen of interest is both structurally and functionally intact.

### 3.3. Stability

Conformational stability and structural integrity of the immunogen must be maintained in order for a vaccine to be stably formulated and administered. Therefore, the effects of different potential storage conditions and stress conditions should be examined in advance of undertaking significant preclinical studies. For an E1E2-based vaccine candidate, conformational stability can be reliably probed by testing conformation-sensitive mAb binding. In general, a useful approach to this type of study is to perform a temperature-dependent binding analysis with a series of mAbs to determine which mAb (or mAbs) are most sensitive to changes in temperature. This mAb (or mAbs) then becomes the diagnostic indicator for a loss of conformational stability. One scenario that is informative is a simple incubation for a defined length of time (e.g., one hour) at varying temperatures (25, 37, 56 °C) to mimic a temporary loss of temperature control. An example of using ELISA for this kind of experiment is shown in Figure 10A. From Figure 10A, it is apparent that individual epitope-binding mAbs are relatively insensitive to this temperature range, whereas the E1E2-specific mAbs, AR4A and AR5A, exhibit a significant loss of binding at 56 °C and are, therefore, good diagnostic indicators for such a study. Another scenario to be investigated in this manner is extended storage at various temperatures, ranging from 4 to 40 °C. Finally, the effects of repeated freeze–thaw cycles on conformational integrity should be examined using similar techniques.

An additional metric for structural integrity is aggregation. While aggregates are often highly immunogenic, unintentional aggregation of an antigen under study fundamentally changes its immunogenic properties and should, therefore, be avoided. The simplest way to assess aggregation is via dynamic light scattering (DLS). DLS is highly sensitive to particle size in solution and, therefore, can detect even minuscule percentages of aggregates. A useful approach to this type of assessment is to perform DLS experiments of antigens prior to and after formulation and subsequent storage under both normal and stress conditions. The threshold of aggregate that can be tolerated is arbitrary (and ideally 0%), but a threshold of 2–5% is reasonable. An example of two samples, one that was properly stored and one that was not, thus exhibiting significant aggregation, are shown in Figure 10B.

## 4. Novel Strategies for Scaffolded E1E2

The native E1E2 complex is not well understood, creating a significant gap in our knowledge regarding HCV entry into cells and its recognition by the immune system, thereby hampering vaccine development. In particular, little information exists regarding how key antigenic regions are presented in the context of the native E1E2 complex. Experimental data describing the native presentation of these regions will shed light on recognition by the immune system and, thereby, advance the design of an HCV vaccine. Similarly, while the region of E2 that interacts with CD81 has been identified, structural characterization of the interaction between E1E2 and CD81 has proven challenging, and, thus, little is known about the interaction. A robust biochemical and structural examination of this interaction will shed light on viral entry. One root cause of this paucity of data characterizing E1E2 is the technical difficulty in producing the membrane-bound complex in sufficient amounts and with sufficient homogeneity (e.g., see Figure 5, Figure 6 and Figure 7 and Figure 9). Liberating the complex from the membrane in its native form would alleviate this bottleneck and allow rigorous determination of the biochemical properties and intermolecular interfaces that define a native E1E2 complex. Development of the soluble native envelope glycoproteins SOSIP (for HIV) [99], DS-Cav1 (for RSV) [100], and the spike extracellular domain for MERS-CoV [101] generated significant advances, indicating that this type of approach has the potential to similarly advance the HCV vaccine development field. Despite the use of E1E2 as an antigen in human clinical trials and nonhuman primate studies [4], beginning in 1994 [28], it has shown limited efficacy [167] and has proven remarkably difficult to purify and characterize in soluble form. In contrast, soluble forms of the truncated core E2 ectodomain can be readily isolated and have led to major insights into E2 structure and immunogenicity [26,35,55,57,58,147,168]. One complicating factor for the HCV E1E2 complex is that coexpressed E1 and E2 ectodomains alone do not form a stable complex [105], indicating that a replacement for the TMDs is required. We discuss recent advancements in creating soluble versions of E1E2 below.

### 4.1. A Covalently Tethered sE1E2 Complex

The earliest successful sE1E2 complex was developed by simply deleting the TMDs and replacing them with flexible linkers between the E1 and E2 ectodomains [169]. These sE1E2 complexes retained a level of recognition by a number of human mAbs, including IGH526, AR3A, and AR5A. Preparations of the tethered sE1E2 chosen for further study exhibited significant heterogeneity, with most of the protein present as dimers, trimers, and higher-order oligomers, as determined by nonreducing SDS-PAGE. Only approximately 5% of the purified tethered sE1E2 was present in the expected monomeric form, indicating significant spurious intermolecular disulfide bonding. Moreover, the tethered sE1E2, when used as a DNA immunogen to immunize mice, elicited only a weakly neutralizing antibody response against either homologous or heterologous strains. However, this work pointed the way towards developing soluble E1E2 constructs as bona fide vaccine candidates.

### 4.2. An sE1E2 Complex Using a De-Novo-Designed Heterodimeric Tag

Another group employed a pair of de-novo-designed helical hairpins as a scaffold to replace the TMDs of E1 and E2 [105]. These helical hairpins, called DHD15 (RCSB ID 6DMA [170]), form heterodimers and are thermally stable. The DHD15-sE1E2 complex was expressed and purified from both insect and mammalian cells. As with other soluble and membrane-associated E1E2 preparations, the DHD15-sE1E2 complex exhibited high molecular weight oligomers in nonreducing SDS-PAGE. In addition, the DHD15-sE1E2 complex eluted from SEC in two peaks, one consistent with a heterodimer and another consistent with oligomeric species. The DHD15-sE1E2 complex was recognized by the conformation-sensitive mAbs IGH526 and AR3A, indicating the preservation of the structural integrity of those epitopes. Binding to several HCV coreceptors was also confirmed. However, binding to conformation-dependent E1E2-specific mAbs like AR4A and AR5A was not demonstrated in that study. Finally, a negative stain EM reconstruction of the sample provided a low-resolution molecular envelope consistent with a heterodimeric complex containing the DHD15 scaffold and two ectodomains. This work established that a scaffolded approach can yield a stable E1E2 heterodimer.

### 4.3. An sE1E2 Complex Using a Heterodimeric Coiled-Coil Scaffold

Recent work by our group explored using linkers and scaffolds to design a native-like sE1E2. Some examples are shown in Figure 11. The simple linker (Figure 11A) exhibited suboptimal antigenicity, particularly with E1E2-specific mAbs, and expressing E1 and E2 as a polyprotein without a linker or scaffold (Figure 11B) failed to yield any secreted E1E2 [22], in contrast with a scaffold that employed the Fos/Jun leucine zipper as a functional replacement for the TMDs (Figure 11C). As in the other two studies, we found that the construct, sE1E2.LZ, forms an intact E1E2 heterodimer. To assess the size and heterogeneity of the complex, we employed the analytical methods described above. AUC for sE1E2.LZ showed two prominent peaks between sedimentation coefficient (S) values 4.9 and 7.7 (Figure 12A), which are approximately consistent with a monomer and dimer of the sE1E2.LZ heterodimer, respectively, and resemble what we have observed in a nonreducing Western blot ([22], not shown). This distribution shows that sE1E2.LZ exhibits less heterogeneity than mbE1E2 (e.g., see Figure 6). Though sE1E2.LZ is not a uniform single species, it is a less complex mixture of E1E2 assemblies than mbE1E2. SEC-MALS analysis largely confirmed the observations from AUC. When compared with standards and analyzed by light scattering, sE1E2.LZ exhibited a single peak in SEC-MALS with an estimated molecular weight at the peak center of 173 kDa, corresponding approximately to a dimer of the sE1E2.LZ heterodimer (Figure 12B). This estimated size is generally consistent with the observed AUC peak around 7.7 S, though the breadth of the peak in SEC-MALS still suggests that sE1E2.LZ displays some heterogeneity in size, corresponding to 1–2 sE1E2.LZ heterodimers, in accordance with the two major peaks from AUC measurements and in contrast to the broader distribution observed for mbE1E2 in Figure 7. Thus, in assessments by multiple analytical techniques, sE1E2.LZ forms a moderately heterogeneous mixture that is nonetheless smaller and closer to the expected size than mbE1E2, representing a potentially improved immunogen for HCV vaccine development. Preservation of native structural elements was confirmed by high-affinity binding to a panel of conformation-specific monoclonal antibodies (Table 2) from all antigenic domains of E2, as well as two neutralizing antibodies specific to native E1E2 (AR4A and AR5A) and its primary receptor, CD81. Immunization of mice with sE1E2.LZ elicited a robust immune response, with high overall antibody titers and neutralization titers against the homologous (H77) strain that were equivalent to those in sera from mice immunized with mbE1E2. This study established the soluble heterodimeric coiled-coil as a bona fide functional replacement for the E1 and E2 TMDs.

## 5. Conclusions and Future Challenges

Recent advances in the production of E1E2 and the structural characterization of E2-antibody complexes have significantly advanced the potential for developing an E1E2-based HCV vaccine. However, major challenges still remain. In particular, mbE1E2 preparations are produced at relatively low yields. Advances in the design of sE1E2 complexes will hopefully alleviate this issue by allowing the use of higher yield expression systems. E1E2 preparations, whether membrane-bound or soluble, exhibit a great deal of heterogeneity, which could hamper advancement through the preclinical and clinical stages. Our sE1E2 complex, while not a single uniform species, is less heterogeneous than mbE1E2 preparations, indicating that liberating E1E2 from the membrane is a step in the right direction. It is possible that cellular quality checks on the secreted complex will result in a more homogeneous preparation. In particular, replacement of the membrane anchor by a soluble scaffold makes the sE1E2 complex an obligate client of the HRD1 pathway of ER-associated degradation (ERAD), a major cellular quality control mechanism that removes terminally misfolded proteins [172]. This particular ERAD pathway, in addition to the E3 ubiquitin ligase HRD1, also requires the cargo receptor SE1L and one of two lectins (either OS-9 or XTP3-B). By contrast, mbE1E2 would be able to access pathways that utilize other E3 ubiquitin ligases and their associated factors [172], and the mbE1E2 extracted from the membrane is likely to be a mix of proteins at various stages of the quality control pathways. Perhaps the routing of sE1E2 through a single quality check pathway, along with the requirement that the pathway be completed in order to secrete sE1E2 from cells, restricts the number of species in solution. Further research in this area would shed light on how to produce more uniform sE1E2 preparations. Moreover, glycosylation and its effects on solution properties, antigenicity, and immunogenicity are not well-understood. More research in this area is sorely needed. The best means of overcoming the heterogeneity issue is a rigorous biophysical analysis of the solution properties of candidate E1E2 vaccine antigens, allowing the selection of designs/modifications that trend towards greater homogeneity. Complementary assessments of antigenicity, receptor binding, and stability will ensure that the drive towards homogenous antigen preparations does not come at the expense of the critical properties required to elicit a robust immune response.

## Figures and Tables

**Figure 1 viruses-13-01027-f001:**
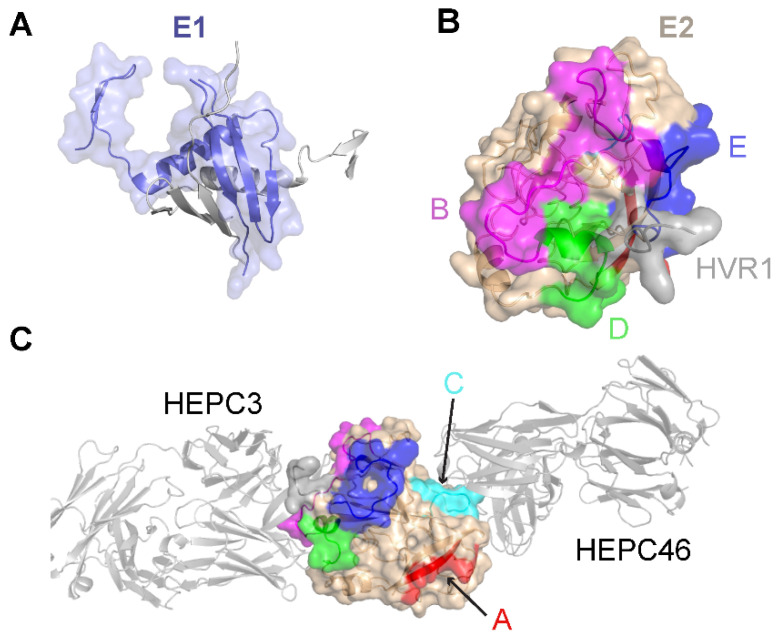
Structures of the E1 and E2 glycoproteins. (**A**) The structure of a truncated E1 glycoprotein ectodomain (PDB: 4UOI), with resolved residues encompassing positions 192–270. An E1 monomer (slate; surface and cartoon representation) is shown with a second E1 monomer (gray) representing the dimeric partner from the X-ray structure assembly. (**B**) The truncated E2 glycoprotein ectodomain structure (PDB: 6MEJ), encompassing E2 residues 405–645, showing the E2 neutralizing face and antigenic domains B, D, and E. Representative residues from the antigenic domains are colored as indicated by the letters, and resolved residues from hypervariable region 1 (HVR1) (residues 405–410) are colored gray. Other residues in E2 are colored tan. (**C**) The same E2 glycoprotein structure as in (**B**), reoriented to show antigenic domains A and C (colored by representative residues and labeled). Bound antibodies HEPC3 (antigenic domain B antibody) and HEPC46 (antigenic domain C antibody), both in gray cartoon, are shown for reference. The neutralizing face is colored as in panel (**B**).

**Figure 2 viruses-13-01027-f002:**
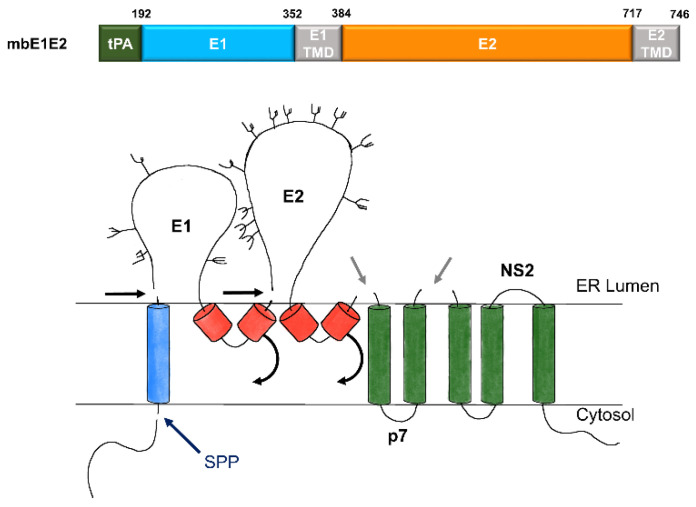
Expression and processing of mbE1E2. (top) Schematic of the E1E2 signal peptide (SP) plus polyprotein expression construct. (bottom) Processing pathway for the N-terminal portion of the HCV polyprotein. Signal peptidase cleavages release E1 and E2 (black and gray arrows). Signal peptide peptidase (SPP) cleavage of HCV core is indicated by a dark blue arrow. Repositioning of E1 and E2 transmembrane domains (TMDs) is indicated by curved arrows. Glycans attached to E1 and E2 are depicted as branched structures. Adapted from [74].

**Figure 3 viruses-13-01027-f003:**
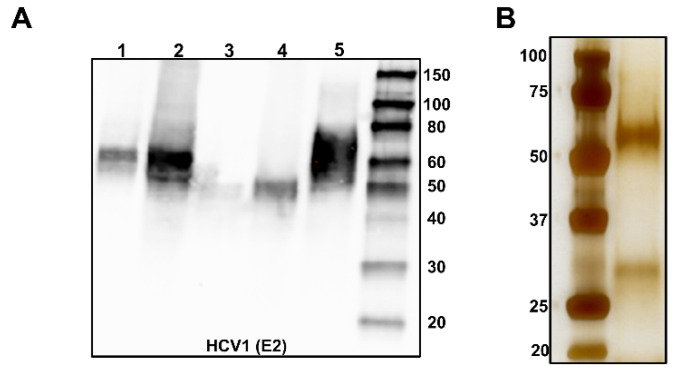
Optimization of expression and purification. (**A**) Relative expression of mbE1E2 in HEK 293F cells (Lanes 1 and 3) versus Expi293F cells (Lanes 2 and 4) analyzed by Western blotting probed with the anti-E2 antibody HCV1. A standard of 250 ng of sE2 is included as a reference (Lane 5). The relative expression levels of the constructs based on quantification of band intensity are 0.26 for Lane 1, 0.81 for Lane 2, 0.09 for Lane 3, 0.31 for Lane 4, and 1.0 for the standard in Lane 5. (**B**) mbE1E2 purified as described in [25] and analyzed by silver-stained SDS-PAGE.

**Figure 4 viruses-13-01027-f004:**
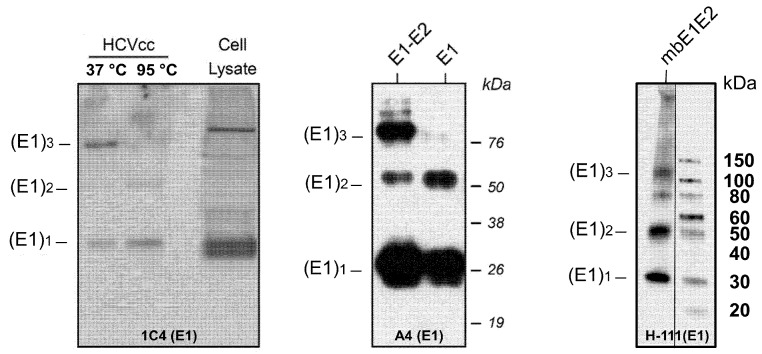
Analysis of HCV E1 trimer formation in different platforms. Nondenaturing (or denaturing, as indicated) Western blot for HCVcc (left), HCVpp (middle), and purified mbE1E2 (right) using the antibodies indicated in each panel. Molecular mass markers (in kilodaltons) are indicated on the right. The oligomeric forms of E1 are indicated on the left. The left and middle panels are reproduced with permission from [116]. In the right panel, duplicate and/or irrelevant lanes between the displayed sample and the marker were deleted for the sake of clarity (indicated by a vertical line).

**Figure 5 viruses-13-01027-f005:**
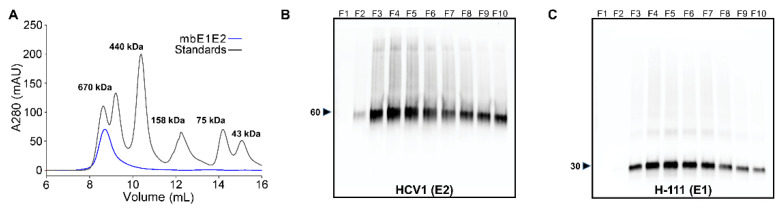
Analysis of purified recombinant mbE1E2. (**A**) SEC profile of mbE1E2 purified from Expi293 cell extracts (**B**) Reducing anti-E2 Western blot analysis of eluted fractions. (**C**) Reducing anti-E1 Western blot analysis of eluted fractions. Molecular weights, in kDa, of the Western blot markers closest to observed bands are indicated on the left panels of (**B**) and (**C**). Reproduced in modified form with permission from [22].

**Figure 6 viruses-13-01027-f006:**
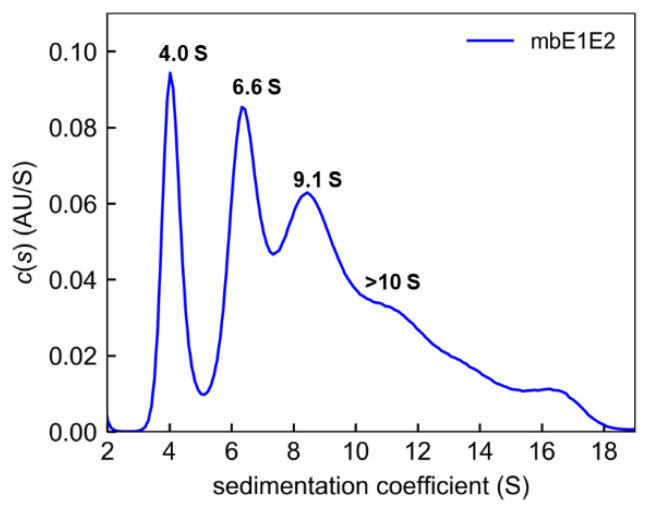
Analysis of mbE1E2 by analytical ultracentrifugation. Shown is the distribution of Lamm equation solutions *c*(*s*) for mbE1E2 (blue). The coefficients for the peaks are shown. Reproduced with permission from [22].

**Figure 7 viruses-13-01027-f007:**
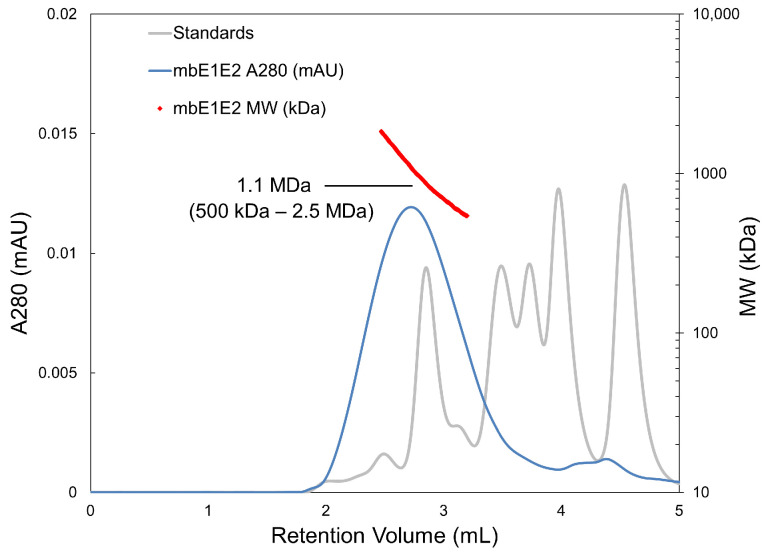
mbE1E2 characterized with SEC-MALS and compared with molecular weight standards. The SEC-MALS chromatograph of mbE1E2 is shown as a blue line. The range of elution volumes within the peak half-maximum is shown as red dots, with the size distribution labeled and enclosed in parentheses. An estimation of molecular weight at the center of the peak is indicated. Molecular weight standards are shown as a grey line corresponding to sizes of 670, 158, 44, 17, and 1.35 kDa. Reproduced with permission from [22].

**Figure 8 viruses-13-01027-f008:**
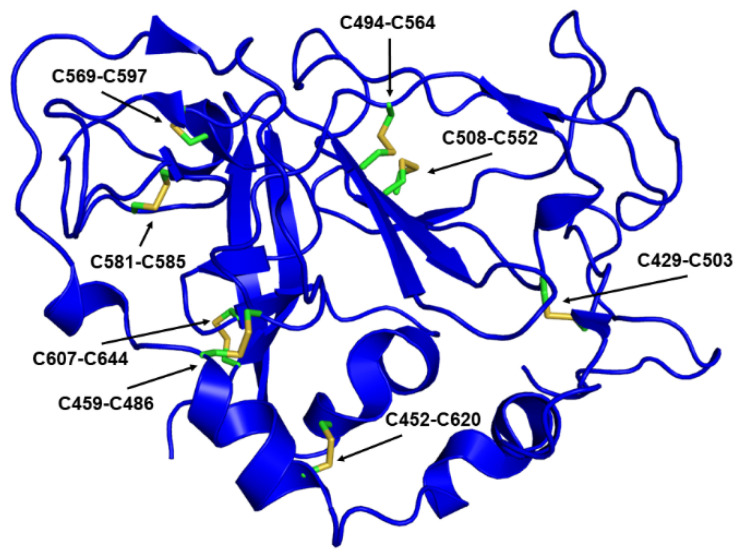
Disulfide bond configuration observed in the E2 ectodomain structure (RCSB ID 6MEI). The numbering shown corresponds to genotype 1b strain 1b09.

**Figure 9 viruses-13-01027-f009:**
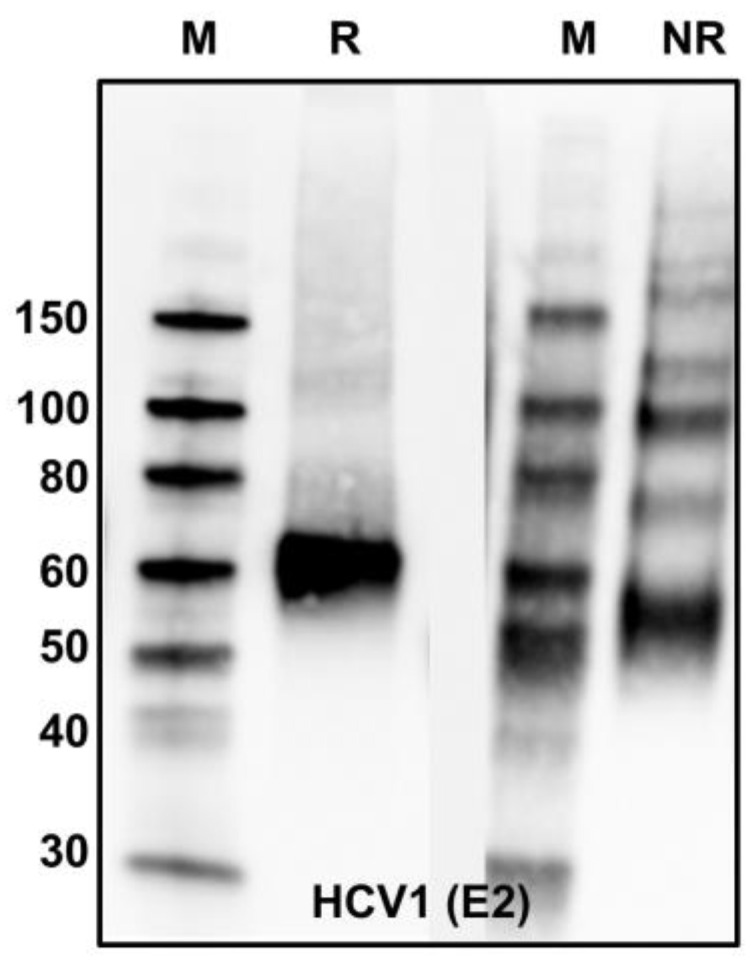
Analysis of purified mbE1E2 by reducing (R) and nonreducing (NR) Western blots. Both blots were probed with the anti-E2 antibody HCV1.

**Figure 10 viruses-13-01027-f010:**
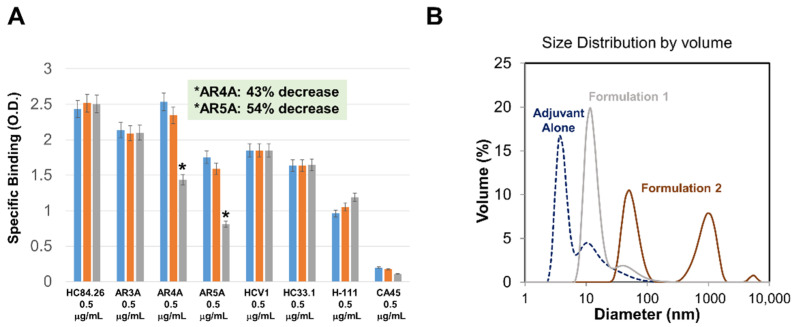
(**A**). Stability analysis of purified mbE1E2 based on temperature-dependent binding to mAbs. Samples were incubated at 25 °C (blue), 37 °C (orange), or 56 °C (gray) for one hour prior to the experiment. (**B**). DLS analysis of mbE1E2 formulated with an adjuvant using two different storage regimens. The size distribution by volume of the adjuvant alone (blue dotted lines) and the two formulations (tan and brown solid lines) is shown.

**Figure 11 viruses-13-01027-f011:**
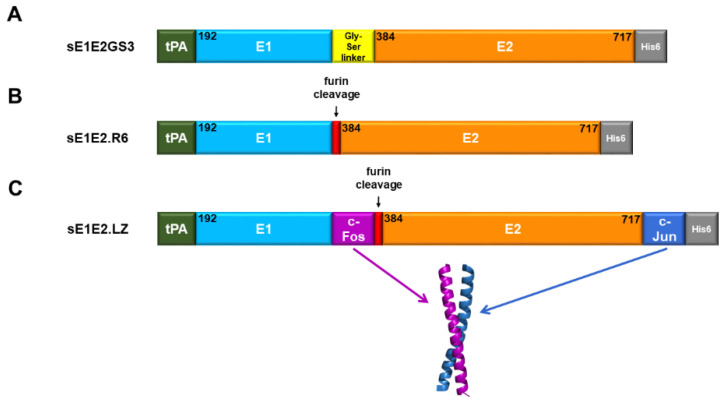
Design of selected sE1E2 constructs. (**A**). Schematic of sE1E2GS3. (**B**). Schematic of sE1E2.R6. (**C**). Schematic of sE1E2.LZ and the X-ray structure of the human c-Fos/c-Jun heterodimer (PDB code: 1FOS); only the coiled-coil region that was used for the sE1E2.LZ scaffold is shown. c-Fos and c-Jun chains were colored to match the diagram of sE1E2.LZ. Regions shown include the tPA signal sequence (green), E1 ectodomain (cyan), E2 ectodomain (orange), Gly-Ser linker (yellow), and c-Fos and c-Jun scaffolds (magenta and blue). Location of the His tag (gray) and furin cleavage site (red) is indicated where applicable. E1E2 residue ranges for each region are noted according to H77 numbering.

**Figure 12 viruses-13-01027-f012:**
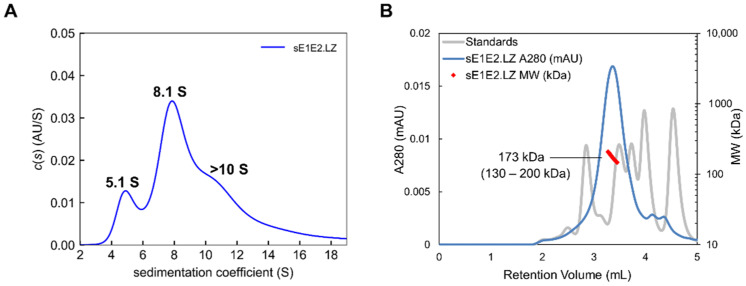
Analytical characterization of sE1E2.LZ heterogeneity. (**A**) AUC profile of purified sE1E2.LZ. Shown is the distribution of Lamm equation solutions *c*(*s*) for sE1E2.LZ (blue line). Calculated sedimentation coefficients for the peaks are labeled. Observed species for sE1E2.LZ approximately correspond to a heterodimer at 5.1 S, a dimer of heterodimers at 8.1 S, and higher-order aggregates at >10 S. (**B**) sE1E2.LZ characterized with SEC-MALS and compared with molecular weight standards. The SEC-MALS chromatograph of sE1E2.LZ is shown as a blue line. A range of elution volumes within the peak half-maximum is shown as red dots, with the size distribution of each range labeled and enclosed in parentheses. An estimation of molecular weight at the center of each peak is indicated. Molecular weight standards are shown as a grey line corresponding to values of 670, 158, 44, 17, and 1.35 kDa. Reproduced with permission from [22].

**Table 1 viruses-13-01027-t001:** HCV comprehensive mAb panel.

Antibody	Species	Antigenic Domain ^1^	Linear (L)/Conformational (C)	Binding Residues ^2^	Neutralizing ^3^	Reference
H-111	Human	E1-Nterm	L	Y192-H202	Y	[154]
IGH526	Human	E1-Cterm	C	H316, M323, M324	Y	[45]
AR1A	Human	A/C	C	W549, Y632	N	[8]
CBH-4B	Human	A/C	C	F627, Y632	N	[41]
AR3A	Human	B	C	L427, F442, W529, D535	Y	[8]
HEPC74	Human	B	C	L427, L438, C452, D535	Y	[155]
HC84.26.WH.5DL	Human	D	C	L441, F442, W616	Y	[7]
HC84.1	Human	D	C	L441, F442	Y	[43]
HC33.1	Human	E	L	Q412-N423	Y	[46]
HCV1	Human	E	L	Q412-N423	Y	[156]
AR4A	Human	E1E2	C	Y201, N205, D698	Y	[39]
AR5A	Human	E1E2	C	Y201, N205, R639, L665	Y	[39]

^1^ E2 antigenic domain (A–E) [153], E1 region, or E1E2 heterodimer target. Antigenic domains A and C are combined due to spatial proximity and overlapping antibody-binding determinants. ^2^ Representative key E1- and/or E2-binding residues from epitope mapping and structural studies, including two global alanine scanning studies [152,153]. H77C sequence and numbering are shown. For linear epitopes, the residue range is shown. ^3^ Y = neutralizing; N = non-neutralizing.

**Table 2 viruses-13-01027-t002:** Binding affinity of mbE1E2, sE1E2.LZ, and sE2 to a panel of monoclonal antibodies measured by dose-dependent ELISA. ^1^

Antibody	Domain ^2^	K_d_ (nM) ^3^			Standard Error (nM)		
		mbE1E2	sE1E2.LZ	sE2	mbE1E2	sE1E2.LZ	sE2
CBH-4D	A	28	26	1	3.2	3.4	0.2
CBH-4G	A	7.8	18	0.5	2.3	3.1	0.3
HC-1 AM ^4^	B	1.5	2.9	3.6	0.06	0.5	0.4
HC-11	B	1.8	3.2	11	0.09	0.4	0.6
CBH-7	C	1	1.7	0.3	0.1	0.1	0.04
HC84.24	D	0.5	1.3	0.7	0.07	0.1	0.1
HC84.26	D	1.2	2.6	0.4	0.03	0.4	0.1
HC33.1	E	3.8	0.9	1.9	0.3	0.09	0.2
HCV1	E	9.8	3.5	6.2	0.3	0.2	0.3
AR4A	E1E2	2.3	16	-	0.2	1.5	-
AR5A	E1E2	1.5	1.7	-	0.2	0.2	-

“-” denotes no binding detected. ^1^ Reproduced with permission from [22]. ^2^ E2 antigenic domain (A–E) [153], E1 region, or E1E2 heterodimer target. ^3^ K_d_, dissociation constant. ^4^ Affinity-matured HC-1 antibody, as previously described [171].

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
