# Peer review of "Structural and Biophysical Characterization of the HCV E1E2 Heterodimer for Vaccine Development"

_viruses, 2021, doi:10.3390/v13061027_

Round 1

Reviewer 1 Report

In this review, Toth and colleagues give a very comprehensive and profound review about the characterization of the HCV E1E2 envelop glycoprotein and how this information can be translated to design a better vaccine against HCV. The authors give a detailed and updated overview of the knowledge accumulated during 30 years of research of HCV envelop glycoprotein, including expression and purification methods and biochemical, biophysical, and structural characterization of the envelope proteins. The biography is up to date and included the latest work published on the subject.

The manuscript is very well designed and written, and it deserved publication. 

I have only a few minor comments:

Minor comments – 

  • In section 3.2.1. the authors give a very informative review about methods that are using to measure antibody binding. It will be beneficial to add the ITC method to this section and write about the advantages (immobilizing free method, thermodynamic parameters, etc.) and disadvantages (need of highly purified soluble antigen, a big amount of sample, etc.). Also, it needs to be mention that ELISA is less commonly used for Kd determination. 
  • In lines 668-672, the authors point that the bnAbs AR4A and AR5A bind to a similar epitope. I think this statement is not so accurate – a competition assay done by Giang et al. (PNAS 2012, the paper that reported the isolation of these bnAbs) indicated that these two bnAbs do not compete.
  • Line 787, please add that the DHD15-sE1E2 complex was not recognized by conformational E1E2 bnAbs.
  • Figure 10A, please increase the size and quality of the figure. It is hard to see the figure labels. And also, the labels (Volume and Diameter) of Figure 10B.    
  • In line 656, please add the following references – Potter, J.A. J. Virol. 2012, 86, 12923-12932; Aleman, F. Proc. Natl. Acad. Sci. U.S.A. 2018, 115, 7569-7574.
  • The format of the references is not uniform – in some references, the title is in “capitalize each word” form, and in others, it is in sentence case format. 
  • Reference 36 and 62 are the same one.

  Typos: 

  • line 68 – an extra “that.”
  • Lines 128, 198 – please fix the “mg.”
  • Reference 44, the title of the paper is in “capitalize all word” format.
  • Line 629, I think you should replace the for with of.
  • Line 754, I believe it should be “and the spike extracellular domain (For MERS-Cov)”.

Author Response

We thank reviewer 1 for the supportive comments and thorough review. In response to the reviewer's concerns, we have done the following:

  1. Added a brief description of the advantaged and disadvantages of using ITC to study antibody-antigen interactions and added the caveat that ELISA is a less commonly-used method for determining dissociation constants.
  2. Revised the section on AR4A and AR5A to state that they don’t compete with each other for binding to E1E2.
  3. Added a line stating that DHD15-sE1E2 recognition by E1E2-specific bnAbs was not shown.
  4. Revised figures 10A and B as requested.
  5. Added the requested references.
  6. Deleted the requested duplicate reference.
  7. Fixed the noted typographical errors.
  8. Regarding the reference title formatting, EndNote inserts the title as entered into the database by the publisher. If I change the format to sentence case, it results in things like E1E2 being changed to e1e2. If I change the format to headline style, E1E2 becomes E1e2. I am just going to leave it alone.

Reviewer 2 Report

Toth et al. presented an interesting comprehensive review entitled " Structural and Biophysical Characterization of the HCV E1E2 Heterodimer for Vaccine Development". The authors discussed the following points in the review a) Importance of E1, E2, and E1E2 epitopes in virus neutralization. And  the structural characterization of E2 antigens and a model of the E1E2 heterodimer.  b) Expression systems for E1E2 and purification steps. In this point the authors discussed in details different expression systems such as CHO cell, HEK293 cells, and BICS expression systems. In addition, purification of mAb E1E2 and their roles in vaccine development.  c)Biophysical characterization of E1E2 including . Size, homogeneity, and oligomeric state and factors affecting size and homogeneity such as disulfides, glycans & methods of purification and analysis of mAb such as size exclusion chromatography (SEC),  analytical ultracentrifugation (AUC), and Size-Exclusion Chromatography Combined with Multiangle Laser Light Scattering (SEC-MALLS). In addition, the authors described Ag-Ab interactions and factors affecting this interaction. (D)  Strategies for scaffolded E1E2 .

In general the review is well designed, nicely flow, comprehensive, and has an acceptable flow. 

Such review is deserved for publication.

Author Response

We thank the reviewer for their supportive comments. Reviewer 2 did not request any changes to the manuscript.